# Moms in motion: Predicting healthcare utilization patterns among mothers in Newfoundland and Labrador

Emily Saunders[1], Noah W. Pevie[1], Shannon Bedford[1], Julie Gosselin[1,2], Nick Harris[1], Joshua A. Rash[1]*

1 Department of Psychology, Memorial University of Newfoundland, St. John's, Newfoundland, Canada,
2 Department of Psychoeducation and Psychology, Université du Québec en Outaouais, Gatineau, QC, Canada

* jarash@mun.ca

**Data Availability Statement:** Data files used in support of this manuscript are available at Open Science Foundation: https://osf.io/ktb56/.

## Abstract

Mothers have a significant influence on family dynamics, child development, and access to family services. There is a lack of literature on the typical Canadian maternal experience and its influence on access to services for mothers despite recognizing the importance of mothers. A cross-sectional study was conducted to address this research gap that employed Andersen's Behavioral Model of Health Service Use in conjunction with a feminist lens. A total of 1,082 mothers who resided in Newfoundland and Labrador (NL) participated in a province-wide survey in 2017 and reported on their wellbeing, family life, and healthcare utilization. Stepwise binomial logistic regressions and linear regressions were used to predict initiation and continued service utilization within the preceding 12 months, respectively. Mothers who participated in this survey were older, and were more likely to be in a relationship than those in the Canadian census, while no difference was observed in annual income. Approximately half of mothers accessed services for themselves over the previous 12-months, with the overwhelming majority accessing services for their children. Medical services were the most likely to be utilized, and mental health and behavioural services were the most likely services to be needed, but not available. Sociodemographic (e.g., age, education attainment), familial relationships and role satisfaction, health need, and health practices predicted maternal initiation and continued use of services, with a larger number of variables influencing maternal service initiation as compared to continuous use of services. Sociodemographic (e.g., maternal age, community population), maternal social support, health need, and maternal health practices predicted maternal access of at least one child service while family relationships, health need, and maternal health practices predicted maternal use of a range of child services conditional on initial access. These results can support the provincial health system to better support access to care by acknowledging the interdependent nature of maternal and child health care utilization. They also highlight the importance of equitable healthcare access in rural locations. Results are discussed in terms of their clinical relevance to health policy.

**Funding:** The authors received no specific funding for this work.

**Competing interests:** The authors have declared that no competing interests exist.

## Introduction

Health services and providers are fundamental determinants of health [1, 2]. Health services include hospital-based programs, emergency services, mental health programs, and home care. Access to these services impact life span and quality of life. The concept of healthcare access is typically measured by healthcare service utilization (HSU) and/or subjective unmet healthcare needs [3–5]. HSU is a measure of actual services received while subjective unmet healthcare needs is the difference between healthcare services deemed necessary to deal with a particular health concern and the actual services received [6]. Measures of HSU capture the use of available services while measures of subjective unmet healthcare needs capture whether the provided care met individuals' needs, and barriers to obtaining care [3].One of the more encompassing HSU frameworks is the Behavioural Model of Health Service Use [7]. This model has received significant empirical support and is the most commonly used framework in research examining healthcare service access and utilization [2–4, 8, 9]. Andersen's conceptual framework was adopted in the present study given the breadth and flexibility of dimensions of HSU, relevance of influences measured, and the significant empirical support for this model (for a review, see [8, 10, 11]).

The Behavioural Model of Health Service Use highlights the role of the healthcare system and social networks in addition to individual and community factors in determining HSU. This model proposes that health outcomes (e.g., perceived health status, evaluated health status, and consumer satisfaction) are determined by: (a) environmental characteristics (i.e., healthcare system and external environment); (b) population characteristics (i.e., predisposing characteristics, enabling resources, and need); and (c) health behaviour (i.e., personal health practices and use of health services). Within the environmental domain, the healthcare system refers to the national health policy, resources, and healthcare organizations while the external environment refers to the economic climate, politics, societal norms, and physical factors [12]. The population characteristics include three factors placed in a sequential order (i.e., predisposing factors to use health services contribute to enabling resources to use health services, which contribute to an individual's needs). Predisposing factors are characteristics of the individual, such as age, sex, ethnicity, social status, education, and cultural/religious beliefs. Enabling resources are system or structural conditions that enable or inhibit use of available services, such as personal and family resources (e.g., health insurance, living arrangements, transportation, family health habits) and community resources (e.g., social supports, distance to health facility, number of providers). Individual needs encompass the complexity and severity of the health condition that leads to the need for health services, and can be viewed from the perspective of the individual seeking care or healthcare provider [12–14]. Personal health practices and HSU are treated as mediating variables of health status and satisfaction rather than the final outcome variables.

Previous research has consistently observed that HSU is higher among females compared to males [13, 15–17]. In an examination of the Canadian Community Health Survey (CCHS) data from 1978 to 2003, researchers observed that females were 40% more likely to see a physician compared to males across all age categories [18]. This relationship has been observed for use of mental health services as well (for a review, see [19]). A recent observational cohort study of healthcare utilization within an integrated healthcare system in Southern California observed that female gender was associated with higher total utilization in individuals diagnosed with COVID [20]. Additionally, studies have identified that single mothers and their children experience health disadvantages [21–24]. For example, researchers utilized the Ontario Health Survey Supplement and observed that lone mothers were significantly more likely to have more mental health problems and use more mental health services compared to

mothers in two-parent families [25]. This relationship persisted after statistically adjusting for income, education, the presence of young children, and maternal age. Multiple theories for sex-based differences in HSU exist, including sex specific conditions (e.g., reproductive care), longer life span among women, differences in health perceptions and symptom reporting, greater likelihood of women seeking preventative care, and physician referral practice patterns [16, 26, 27].

Prevailing gender ideologies and the experiences of mothers is instrumental in understanding access to services [21, 28–30]. The term 'intensive mothering' has been used to describe the hegemonic ideology of motherhood in Western society in which maternal expectations are child-centered, expert guided, emotionally absorbing, and labor intensive [31]. Mothers are expected to be primarily responsible for their children's development, and to prioritize their children's wellbeing and needs over their own needs and convenience [31]. It is expected that mothers take responsibility for monitoring the health of their children, accessing healthcare services for their children, and making decisions regarding the healthcare of their children [30, 32, 33]. As a result, mothers are typically held responsible and disproportionately blamed compared to fathers for the health and wellbeing of their children (i.e., mother-blame) [29]. One setting in which women often experience mother-blame is in formal healthcare, where mothers frequently interact with and are subject to the scrutiny of healthcare providers [29, 34–36]. Despite mothers' experience of mother-blame in formal healthcare settings and the negative impact that the dominant ideology of motherhood can have on women's mental health, little is known about whether this impacts their use of healthcare services for themselves and their children.

The wellbeing of children can impact maternal health and the experience of motherhood. Research on the relationship between maternal-child attachment and health outcomes has suggested that prenatal attachment impacts women's adaptation to the role of motherhood, including facilitating motivation for positive health behaviours and serving as a protective factor against perinatal depression [37, 38]. Previous research also illuminates a positive relationship between maternal prenatal care and HSU for young children [39–42]. These findings suggest that demographic variables, child health, and maternal health are closely linked to maternal wellbeing, indicating the importance of examining these variables to understand the motherhood experience as it related to health service utilization.

Parenting stress is the psychological distress experienced by parents trying to meet parenting role demands [43, 44]. High levels of parenting stress have been associated with adverse outcomes (e.g., depression, anxiety, maladaptive behaviours) among children and parents [45–49] Parenting stress typically increases in parents of children with mental and physical health conditions [44, 50, 51]. A meta-analysis of 547 studies regarding parenting stress in families with a child with a chronic physical condition observed significantly higher maternal stress than paternal stress [33]. Protective factors for parenting stress in these studies included marriage or cohabitation, positive marital quality, and social support [33]. In line with the dominant ideology of 'intensive mothering', higher parental stress experienced by mothers has been hypothesized to be a result of mothers often being the primary caregiver of their children's health [30, 33, 52, 53].

Previous research on motherhood has observed that, while intensive parenting beliefs negatively impact women's mental health, motherhood itself can be associated with increased life satisfaction and meaning [54]. In line with this, research with women experiencing serious mental illness has consistently highlighted the importance and centrality of their role as a mother to their identity. Women described that being a mother gave them purpose and maintaining contact with their children was a priority [55, 56]. In line with the value placed on motherhood, women self-reported that fear of negative evaluation and custody loss negatively

impacted their willingness to seek help or communicate distress with healthcare professionals [57–61].

Little is known about whether mothers are accessing healthcare services that are available to them, and the factors that impact their decision to access services for themselves and their children. In interviews conducted with 379 women, almost half of mothers with serious mental illness reported receiving minimal or no services related to parenting despite a perceived need for services, such as parenting skill training and support. Some evidence suggests that mothers experiencing mental illness following childbirth do receive more healthcare services [62]. Using questionnaires completed by 594 women in British Columbia at 1, 4, and 8 weeks postpartum, researchers observed that women with depressive symptomatology had more contacts with a health professional in their first month postpartum compared to mothers without depressive symptomatology, including their family physician and public health nursing services [63]. Similarly, in a study that examined HSU and satisfaction with services among 574 women 16 weeks after childbirth in Australia, the researchers observed that one in five women scored above clinical cut-offs on the Edinburgh Postnatal Depression Scale, and were more likely to have contact with services (e.g., psychiatrist, social worker, group therapy, paediatrician, or general practitioner [GP]), but were less satisfied with services than mothers without clinically relevant symptoms of depression [64].

Research regarding HSU among children of mothers coping with mental illness is also mixed. A longitudinal study on more than 65,000 Canadian children aged 1–15 years and their parents, found that mothers who reported their health to be poor or to experience depression in previous surveys were more likely to describe their child's current health as poor [65]. Similarly, researchers have found that children were more likely to access medical and mental healthcare if their parents experienced depression. The depressed parent was the mother among 77% of the children who had a depressed parent and was accessing healthcare services [59, 66]. In a systematic review of associations between parent and child mental health with paediatric HSU, parental mental health problems were associated with more child outpatient visits across 13 studies, with the exception of maternal depression [67].

The present study sought to: 1) capture mothers' experience of health, wellbeing, and need for services for themselves and their children; and 2) create a predictive model of mothers' utilization of health services for themselves and their children. The data for this study was obtained from a group of mothers from Newfoundland and Labrador (NL), who participated in the online provincial survey known as 'The Newfoundland Motherhood Project'.

NL is one of the least populated provinces in Canada with a population of 510,550 in 2021 [68]. According to Statistics Canada, as of 2021, 68.57% of census families were married couples, 15.67% were common-law couples, and 15.76% were lone-parent families within NL. Among married and common-law couples in NL in 2021, 82.19% had children [69].NL has a high percentage of mobile and interprovincial employment within the labour force. Mobile work is when there is a significant distance between the place where one works and lives, often resulting in absence from the home for an extended period of time [70]. Interprovincial employment is a specific form of mobile work in which an individual works in one province while maintaining residence in another province [71]. NL and the Yukon have the highest population percentage of interprovincial employment in Canada, with 9.9% of residents of NL working interprovincially in 2011 [71]. Data from Statistics Canada between 2005 and 2014 indicate that the majority of interprovincial employees were married or living common-law and between the ages of 25 to 39, and in 2014 there was an average ratio of 3.5 males to female [72]. While the study did not document the percentage of interprovincial employees that were parents, the large volume of mobile workers within NL, of which the majority are male and in relationships, suggest many mothers in NL experience characteristics of lone-parent families

while their partners are away. This can impact their maternal experience, mental health and wellbeing, and ability to access services.

Within NL, 53% of the population lives in urban areas while 47% lives in rural areas [69]. Regional differences within NL exist in access to healthcare, access to parenting information, and attitudes towards mental health. Mathews and Edwards observed that 15% of NL residents did not have a regular doctor, but 74% of those without a doctor were living in a rural area [73]. Rural residents without a regular doctor were typically female, older, and reported poorer health, while urban and semi-urban residents without a doctor were typically male, well-off financially, and healthy. A more recent survey observed that the percentage of NL residents without a regular doctor has increased to 19% as of 2021 [74]. Consistent with previous research, the survey identified that the number of residents without a regular doctor is higher in rural areas of NL, and among individuals with lower incomes [74].

## Methods and materials

### Participants

Mothers who resided in NL were eligible to participate regardless of age, marital status, sexual orientation, gender identity, or parenting step and/or adopted children. The study primarily focused on mothers with children under the age of 18 years, as many of the survey questions were more applicable for mothers with children who lived in the family home. Participants were consented online and provided electronic consent prior to completing the survey. Ethics approval for secondary use of the data set for this project was obtained from the Memorial University of Newfoundland Interdisciplinary Committee on Ethics in Human Research (ICEHR: 20190723).

### Design and procedure

This study was a cross-sectional quantitative survey design with participants recruited using a non-probability convenience sample. Participants were recruited through advertisement on social media, radio interviews and posters, and information packages placed in day-care centers, community centers and healthcare centers in locations across NL. Targeted recruitment efforts were made to over-sample under-represented groups (e.g., adoptive mothers and mothers living in Labrador) to ensure the sample was inclusive of diverse family structures and mothers across the province.

Participants were asked to complete an online survey housed on Qualtrics. Individualized survey links remained active for seven days. While no one availed of it, participants were given the option to complete the survey over the phone. This survey was modeled after previous surveys titled 'Canadian Survey on Parenthood,' and 'Growing Up in Australia' with slight adjustments to language to reflect the context within NL. The survey consisted of 68 multi-item questions and scales with good psychometric properties that measures maternal and family experience, including validated self-report measures and questions regarding maternal and child demographic variables. Mothers were asked to complete child-parent scales once for each type of child (i.e., biological, step, and adopted) that applied to their parenting situation. Mothers completed the child scales based on their experience with the child who had the next birthday (e.g., a mother with a biological and adopted child completed the scales twice, being prompted each time to respond according to their experience with the biological child who has the next birthday and adopted child who has the next birthday). This method was chosen to reduce completion time, maximize participation, and randomize the response while still getting an accurate depiction of parent-child relationships. Participants were exposed to a minimum of 100 items and a maximum of 205. Participants with no children under the age of 18

and no partner were exposed to less questions than participants with biological, step, and adopted children and multiple partners. On average, the survey took 20–30 minutes to complete. The survey was launched on February 1, 2017 and was closed on October 5, 2017. Data forming the basis of this project was accessed for research purposes in October 2018.

## Measures

**Demographics.** Demographic questions related to maternal, child, and family experiences were included in the study to develop an understanding of the individual characteristics of participants and allow for control of extraneous variables. Specifically, 23 socio-demographic questions regarding maternal age, income, education, occupation, geographic location, and family structure were included. There were 15 socio-demographics questions regarding maternal, family, and child physical health, mental health, family and community social support (e.g., "*How often do you see, talk to, or email the following people?*" and "*In general, how satisfied are you with the social support you are receiving?*"), the experience of adverse events during the previous year (e.g., "*In the last 12 months, have any of the following happened to you or your current partner?*"), satisfaction with healthcare services (e.g., "*overall how satisfied are you with the availability of family health services in your community?*"), need and access to services (e.g., "*In the last 12 months have you used any of these services for your child?*" and "*In the last 12 months have there been any of the services listed that anyone in the family has needed but could not get?*"). Lastly, there were 21 socio-demographic questions related to type of children in the family (i.e., biological, step, adopted), custody arrangement, parenting situation, and child specific questions. These questions were modeled after a Canadian survey examining maternal experiences in stepfamilies, entitled 'Canadian Survey on Parenthood' [75] and from an Australian longitudinal study, entitled 'Growing Up in Australia', which examined the nature and impacts of early family composition, relationships, and dynamics on the adult outcome domains [76].

**Outcome questionnaire– 45.** The Outcome Questionnaire (OQ-45) [77] was used to measure three main areas of mental health: symptom distress, interpersonal relationships, and social functioning. The symptom distress scale consists of 25 items (e.g., "*I feel no interest in things*") assessing symptoms characteristic of mental health disorders (e.g., depressive and anxiety symptoms). The interpersonal relationships scale consists of 11 items (e.g., "*I am concerned about family troubles*") assessing difficulties with family, friend, and marital relationships. The social functioning scale consists of 9 items (e.g., "*I find my work/school satisfying*") assessing problems in employment, education, and leisure activities. Within each subscale, 3 items assess for the presence of positive mental states (e.g., "*I feel loved and wanted*"). All 45 items are rated on a five-point scale, with response options ranging from never (1) to almost always (5). Negatively keyed items were reverse scored, and a total score was computed by summing the item responses. Total scores on the subjective discomfort scale range from 0–100, with higher scores indicating more symptoms and scores 36 and above indicating symptoms of clinical significance. Possible total scores on the scale has been validated in clinical, community, and undergraduate samples and has high reliability and construct validity [77].The OQ-45 has an internal consistency of .93 [78]. In this study the scale had strong internal reliability with a Cronbach's alpha of .94.

**Parental sense of competence scale.** The Parental Sense of Competence Scale (PSOC) [79] was used to measure two dimensions of parental self-esteem and competence: satisfaction and efficacy. The satisfaction subscale assessed the degree to which a parent felt frustrated, anxious, or motivated in their role as a parent and the efficacy subscale assessed the degree to which the parent felt competent, capable of problem solving, and familiar with parenting

duties. The scale consists of 17 items (e.g., "*Being a parent makes me tense and anxious*") rated on a six-point scale, with response options ranging from strongly disagree (1) to strongly agree (6). Negatively keyed items were reverse scored and a total score was computed by summing the item responses. Total scores on the PSOC range from 17–102, with higher scores indicating a larger sense of parental self-esteem and competence. The scale has good internal consistency, with a Cronbach's alpha of .79, and is a valid measure compared to other measures of family functioning and control [80]. The PSOC scale is the most commonly used measure of parental self-efficacy and has been validated in samples including Australian parents and Canadian mothers [80–82]. In this study the total scale had an internal Cronbach's alpha of .53, the efficacy subscale had a Cronbach's alpha of .67, and the satisfaction subscale had a Cronbach's alpha of .78.

**Child-parent relationship scale: Short form.**   The Child-Parent Relationship Scale (CPRS) [76, 83] was used to measure two dimensions of the relationship between the mother and child: conflict and closeness. The conflict subscale assessed the degree to which a parent felt negativity and hostility in the parent-child relationship and the closeness scale assessed the degree to which a parent felt warmth and affection in the parent-child relationship. The scale consists of 15 items (e.g., *"This child and I always seem to be struggling with each other"*) rated on a six-point scale, with response options ranging from definitely does not apply (1) to definitely applies (6). The conflict subscale consists of eight items and total scores on this scale range from 8–48, with higher scores indicating greater perceived conflict in the parent-child relationship. The closeness subscale consists of seven items and total scores on this scale range from 7–42, with higher scores indicating a stronger perceived relationship between the parent and child. This scale has strong internal consistency, with a Cronbach's alpha of .84 for the parental conflict subscale and a Cronbach's alpha of .69 for the parental closeness subscales [77, 84]. Participants were asked to complete this scale once per each type of child that applied to their parenting situation (i.e., biological, step, and/or adopted). In this study the total scale had an internal Cronbach's alpha of .53, the biological CPRS conflict subscale had a Cronbach's alpha of .91 and the closeness subscale had a Cronbach's alpha of .63.

**Quality of co-parental communication scale.**   The Quality of Co-Parental Communication Scale (QCPCS) [85] was used to measure mother's perception of the quality of communication related to child rearing issues with their parenting partners. This scale measures two dimensions of co-parental communication: conflict and support. The conflict subscale assessed the degree to which a parent felt negativity and hostility in communicating with a co-parent and the support scale assessed the degree to which a parent felt there was parental alignment and positive communication with the co-parent on child rearing issues. The scale consists of 10 items (e.g., *"When you and your co-parent discuss parenting issues, how often does an argument result?"*) rated on a five-point scale, with response options ranging from never (1) to always (5). The conflict subscale consists of four items and total scores on this scale range from 4–20, with higher scores indicating greater perceived disagreement between co-parents. The support subscale consists of six items and total scores on this scale range from 6–30, with higher scores indicating greater perceived support and agreement between co-parents. The scale has good internal consistency, with a Cronbach's alpha of .88 for parents and .84 for step-parents, and has high inter-rater reliability between co-parents [86]. In the present study, the Cronbach's alpha was .45 for the total scale, .83 for the conflict subscale, and .84 for the conflict scale. Participants were asked to complete this scale once for each co-parenting situation (i.e., current partner and/or former partner).

**Kansas family life satisfaction scale.**   Maternal family life satisfaction was measured using the Kansas Family Life Satisfaction Scale [87]. This scale consists of four items assessing satisfaction within family life. For the purpose of this study, only the three items of the scale

pertaining to maternal experience were included. The three items (e.g., "*How satisfied are you with your partner*") were rated on a 7-point scale, with response options ranging from extremely dissatisfied (1) to extremely satisfied (7). The scale has adequate internal consistency, reliability, and validity [80]. In the present study, the total scale had a Cronbach's alpha of .61.

## Data cleaning and handling of missing data

Casewise deletion was used to omit participants with significant missing data from the primary database, defined as cases where less than 50% of the survey was completed. This strategy ensured unbiased estimates and conservative results [88, 89]]. Visual inspection of missing data showed that participants with more than 50% of missing data did not complete the survey and missing data was due to attrition rather than skipping questions throughout the survey. Additional cases were deleted when there was missing data for more than 10% of demographic items due to the relevance of demographic data for statistical analyses.

Patterns of missing data were evaluated in the final dataset. Items not shown to participants due to skip-logic were coded as "not applicable" and represented numerically with "888". Items shown to participants and left blank were coded as "missing" and represented numerically by "999". Little's test of Missing Completely at Random (MCAR) was non-significant, indicating data among the 1,082 participants included in the final sample were missing completely at random, $x^2$ = 999.784, DF = 2264, $p$ = 1.00. Missing data were imputed using expectation maximization within the missing values analysis package in SPSS. See Table 1 for frequency of missing data per item. In this sample, the number of mothers who reported having step-children, adopted children, a combination of biological, adopted, and step-children, and children above the age of 18 were grouped together due to the small number of mothers who endorsed each of these family structures. Additionally, only co-parental communication (QCPCS) with current partners and child-parent relationship (CPRS) with biological children were included in all statistical analyses due to lack of power for other types of relationships (e.g., QCPCS with former partner).

## Statistical analysis

Statistical analyses were conducted using IBM SPSS version 27 [90]. The cleaned database was screened for accuracy, outliers, and normal distribution of variables. Non-normally distributed variables were log-transformed to create a new normalized variable to be used for statistical testing. Descriptive statistics were performed to characterize the sample. Frequency and counts were used to summarize discrete variables. Means and standard deviations were calculated to summarize continuous variables. Unless otherwise stated, $p < .05$ was considered statistically significant result.

**Representativeness of the current sample to Mothers across NL.** Bivariate analyses were completed to compare the current sample of NL mothers to the Canadian census data available through Statistics Canada to assess for representativeness and generalizability of the motherhood survey. Data on the mean age of mothers in NL was available from the 2013 Canadian Census and data from the 2016 Canadian Census on the marital status of women in NL and median household income was available [91, 92].

**Predicting Mothers' utilization of services for themselves and their children.** Andersen's theoretical framework delineates multiple dimensions of access to care, including initiated and continued access. Consistent with this characterization of access, previous literature has shown different factors impact adults initial use of healthcare compared to their continuous use of healthcare services once they have accessed the healthcare system [18]. As such,

**Table 1. Sociodemographic and health data for the sample who completed the Motherhood survey.**

| Characteristics | N | % |
|---|---|---|
| Race/ethnicity | *1082* | |
| Caucasian | 1029 | 95.2 |
| Native North American | 29 | 2.7 |
| Asian | 3 | 0.3 |
| Other | 20 | 1.8 |
| Age (*M* = 34.66, *SD* = 8.06) | *1082* | |
| 18–25 | 77 | 7.1 |
| 26–33 | 414 | 38.3 |
| 34–40 | 322 | 29.8 |
| 41–47 | 173 | 16.0 |
| 47–55 | 75 | 6.9 |
| Over 55 | 21 | 1.9 |
| Population Centre | *1082* | |
| Provincial capital region | 407 | 37.6 |
| City | 172 | 15.9 |
| Town | 238 | 22.0 |
| Small town | 233 | 21.5 |
| Labrador | 32 | 3.0 |
| Household Income (*M* = 6.57, *SD* = 2.57) | *1082* | |
| 0–14,999 (1) | 29 | 2.7 |
| 15,000–29,999 (2) | 78 | 7.2 |
| 30,000–44,999 (3) | 79 | 7.3 |
| 45,000–59,999 (4) | 76 | 7.0 |
| 60,000–74,999 (5) | 81 | 7.5 |
| 75,000–89,999 (6) | 95 | 8.8 |
| 90,000–99,999 (7) | 110 | 10.2 |
| 100,000–149,999 (8) | 286 | 26.4 |
| 150,000–199,999 (9) | 155 | 14.3 |
| 200,000–249,999 (10) | 58 | 5.4 |
| 250,000+ (11) | 33 | 3.0 |
| Missing Data | 2 | 0.2 |
| Education (*M* = 5.30, *SD* = 1.890) | *1082* | |
| Some High School (1) | 13 | 1.2 |
| High School Diploma (2) | 50 | 4.6 |
| Some College (3) | 72 | 6.7 |
| College Diploma (4) | 330 | 30.5 |
| Some Undergraduate studies (5) | 76 | 7.0 |
| Undergraduate Degree (6) | 320 | 29.6 |
| Some Masters studies (7) | 46 | 4.3 |
| Master's Degree (8) | 142 | 13.1 |
| Some Doctoral studies (9) | 9 | 0.8 |
| Doctoral Degree (10) | 15 | 1.4 |
| Some Post-Doctoral studies (11) | 1 | 0.1 |
| Post-Doctoral Degree (12) | 7 | 0.6 |
| Missing Data | 1 | 0.1 |
| Currently Working | *1082* | |
| Yes | 738 | 68.2 |

(*Continued*)

**Table 1.** (Continued)

| Characteristics | N | % |
|---|---|---|
| No | 344 | 31.8 |
| Current Occupation | *738* | |
| Professional | 320 | 43.3 |
| Office Employee | 264 | 35.8 |
| Tradesperson | 22 | 3.0 |
| Self Employed | 37 | 5.0 |
| Student | 4 | 0.5 |
| Stay at Home Parent | 5 | 0.7 |
| Other | 86 | 11.7 |
| Mobile Relationship | 1082 | |
| Yes | 443 | 40.9 |
| No | 639 | 59.1 |
| Marital Status | *1082* | |
| Married | 689 | 63.7 |
| Cohabiting | 192 | 17.7 |
| Single, never married | 81 | 7.5 |
| Separated/divorced | 59 | 5.5 |
| Remarried after divorce/separation | 20 | 1.8 |
| Cohabiting after divorce/separation | 33 | 3.0 |
| Widowed | 5 | 0.5 |
| Remarried after being widowed | 3 | 0.3 |
| Sexual Orientation | *1082* | |
| Heterosexual | 1028 | 94.9 |
| Homosexual | 6 | 0.6 |
| Bisexual | 41 | 3.8 |
| Asexual | 1 | 0.1 |
| Transgender/transsexual | 2 | 0.2 |
| Other | 4 | 0.4 |
| Type of Child(ren) | *1082* | |
| Biological | 956 | 88.2 |
| Stepchild | 3 | 0.3 |
| Adopted child | 13 | 1.2 |
| Biological and step | 51 | 4.7 |
| Biological and adopted | 4 | 0.4 |
| No children under age 18 | 54 | 5.0 |
| Missing Data | 1 | 0.9 |
| Experience of Major Event (past 12 months) | *983* | |
| Birth of a child or pregnancy | 277 | 28.2 |
| Serious illness, injury, or assault to you/partner | 81 | 8.2 |
| Serious illness, injury, or assault to close relative | 115 | 11.7 |
| Parent, partner, or child has died | 38 | 3.9 |
| Close family friend or relative has died | 254 | 25.8 |
| Separated from spouse/partner | 43 | 4.4 |
| Broken off a steady romantic relationship | 28 | 2.8 |
| Started living with a new partner | 15 | 1.5 |
| Someone new moved into household | 52 | 5.3 |
| Serious problem with family/friend/neighbour | 164 | 16.7 |

(*Continued*)

**Table 1.** (Continued)

| Characteristics | *N* | % |
|---|---|---|
| Had a major financial crisis | 183 | 18.6 |
| Crisis or major disappointment at work | 193 | 19.6 |
| Thought you would soon lose your job | 143 | 14.5 |
| Lost your job, not by choice | 107 | 10.9 |
| Sought work unsuccessfully for 1 month or longer | 152 | 15.5 |
| Problems with police or court appearance | 17 | 1.7 |
| Had something you value lost or stolen | 30 | 3.1 |
| Someone in household had alcohol or drug issue | 46 | 4.7 |
| Changed job or returned to work | 278 | 28.3 |
| Maternal Mental Health Diagnosis | 1082 | |
| Yes | 347 | 32.1 |
| No | 713 | 65.9 |
| Prefer not to say | 20 | 1.8 |
| Missing | 2 | 0.2 |
| Mental Health Diagnosis | 347 | |
| Anxiety | 81 | 23.5 |
| Depression | 89 | 25.8 |
| Anxiety and Depression | 117 | 33.9 |
| Other Mental Health Diagnosis | 22 | 6.4 |
| Multiple Diagnosis | 36 | 10.4 |
| Missing Data | 3 | 0.9 |

*Note.* Other mental health diagnoses include posttraumatic stress disorder, Borderline Personality Disorder; Bipolar Disorder (BPD), BP, Eating Disorder, Attention Deficit Hyperactivity Disorder, Obsessive Compulsive Disorder. Multiple Diagnosis is defined as 2 or more diagnosis, excluding anxiety and depression.

separate statistical models were performed to evaluate predictors of service initiation and continuous service use. Specifically, stepwise binomial logistic regressions and linear regressions were completed to build predictive models for initiation and continuous use of services, respectively. Variables supported by the literature to be potentially related to health service utilization for adults and children were included in the multivariate analyses.

**Predicting Mothers' adult service initiation.** To determine the unique contribution of environment factors, population characteristics (i.e., predisposing characteristics, enabling resources, need), and health behaviours in predicting the likelihood of mothers accessing at least one adult health service in NL in the previous 12 months, a stepwise binomial logistic regression was performed. The criterion variable (i.e., dependent variable) was access to adult services coded in a binary manner (0 = none, 1 = at least one). Variables included in STEP1 were forced entered into the regression to statistically adjust for these variables as co-variates. Predictor variables included in STEP2-5 (i.e., independent variables) were entered using a forward stepwise approach.

Based on Andersen's theoretical framework for understanding access to services, the following STEPs were chosen for the regression: 1) Environment and predisposing characteristics as measured by population centre, geographical location (i.e., urban or rural), maternal age, maternal education attainment, maternal occupation status (i.e., working or not working), household income, marital status, type of child (i.e., only biological or other), and availability of adult services operationalized as a perceived need for adult services that were unable to be accessed in the previous 12 months; 2) enabling resources as measured by satisfaction with

social support, frequency of contact with social support, mobile relationships, KFLS total score, biological CPRS-closeness, biological CPRS-conflict, current partner QCPCS-support, current partner QCPCS-conflict, PSCS-satisfaction, PSCS-efficacy, and satisfaction with availability of adult services; 3) need characteristics as measured by self-reported mental health diagnosis, child's perceived overall health status, mother's perceived overall health status, and OQ total score; and 4) health behaviours as measured by total child services accessed in the previous 12 months, weekly produce consumption, weekly exercise frequency, and sleep quality. Prior to conducting the analysis, assumptions were checked to ensure the data met the criteria for the analysis. Non-normally distributed variables were log-transformed to create a new normalized variable to be used for statistical testing. There was no evidence of multicollinearity, as assessed by tolerance values greater than 0.1, or significant outliers.

**Predicting Mothers' child service initiation.** A stepwise binomial logistic regression was also performed to determine the unique contribution of environment factors, population characteristics (i.e., predisposing characteristics, enabling resources, need), and health behaviours in predicting the likelihood of mothers accessing at least one child health service in NL in the previous 12 months. This stepwise binomial logistic regression model was a replicate of the model used to predict access to at least one adult service with three exceptions: 1) the criterion variable (i.e., dependent variable) was access to child services coded in a binary manner (0 = none, 1 = any); 2) the variable measuring total child services accessed in the previous 12 months was not included in STEP4; and 3) no variable measuring availability of adult or child services in the previous 12 months was included in STEP1 as a covariate. There was no evidence of multicollinearity, as assessed by tolerance values greater than 0.1, or significant outliers.

**Predicting Mothers' continuous utilization of adult services.** A linear regression was performed to determine the unique contribution of environment factors, population characteristics, and health behaviours in predicting the likelihood of mothers continued use of a range of services for themselves in NL in the previous 12 months. In this analysis only mothers who had accessed at least one service in the previous 12 months were included to better understand the likelihood of mothers accessing a range of services once they initiated access to the healthcare system within the previous 12 months. Variables included in STEP1 were forced entered into the regression to statistically adjust for the variables as co-variates. Variables included in STEP2-5 (i.e., independent variables) were entered using a forward stepwise approach.

This linear regression model was a replicate of the logistic regression model used to predict initiation of adult services with one exception: 1) the criterion variable (i.e., dependent variable) access to adult services was coded in a continuous manner. All other aspects of this analysis remained the same to the logistic regression performed to assess the likelihood of mothers accessing at least one adult service in NL in the previous 12 months. Linearity was assessed by partial regression plots. There was independence of residuals, as assessed by a Durbin-Watson statistic of 1.916 for adult services accessed. Homoscedasticity was assessed by unstandardized predicted values. There was no evidence of multicollinearity, as assessed by tolerance values greater than 0.1.

**Predicting Mothers' continuous utilization of child services.** A linear regression was performed to determine the unique contribution of environment factors, population characteristics, and health behaviours in predicting the likelihood of mothers continued use of a range of services for their children in NL in the previous 12 months. This linear regression model was a replicate of the logistic regression model used to predict initiation of child services with one exception: 1) the outcome variable was access to child services coded in a continuous manner. There was independence of residuals, as assessed by a Durbin-Watson statistic of

1.718 for child services accessed. Homoscedasticity was assessed by unstandardized predicted values. There was no evidence of multicollinearity, as assessed by tolerance values greater than 0.1.

## Results

### Demographic characteristics

A sample of 1,450 mothers who live in NL took part in the study, and 1,082 completed at least 90% of the questionnaire. The age of mothers in the present study ranged from 18 to 55+, with a mean age of 34.66 (SD = 8.06) years. The majority of participants were Caucasian (95.2%), heterosexual (94.9%), married for the first time (63.7%), and had only biological children (88.2%). The median household income interval reported was between $75, 000 and $89, 999 CAD and the majority of mothers reported being currently employed (68.2%). Of the mothers currently working, 43.3% reported employment as a professional and 35.8% as an office employee. Almost half (40.9%) of the sample identified being in a mobile relationship. See Table 1 for more information on socio-demographic information.

### Representativeness of the current sample to Mothers across NL

The average age of mothers in NL who completed the 2013 Canadian census was 28.9 years. A one sample t-test, $t(1081) = 27.584$, $p < .001$, $d = .84$ indicated that the average age of mothers in the Motherhood survey (M = 35.66, SD = 8.06) was significantly older than the average age of mothers in the Canadian census by 2 mean of 6.76 years. A chi-square goodness of fit test indicated that the participants in the Motherhood survey had a different distribution of marital status compared to the NL census data, $\chi^2$ (4, $N = 1082$) = 381.58, $p < .001$. The Motherhood survey had a significantly higher proportion of married (63.7%) and cohabiting (17.7%) mothers, and a significantly lower proportion of single, never married (7.5%), separated or divorced (5.5%) and widowed (0.5%) mothers compared to the 2016 census data, in which 50.25% of women were married, 10.15% were cohabiting, 21.83% were single, never married, 7.62% were divorced and 10.15% were widowed. Lastly, the 2016 Canadian census reported the median household income in NL was 83, 589 [92] which is within the same range as the median household income identified in the present study (75, 000 and 89, 999).

### Health and wellbeing indicators

The majority (89.8%) of mothers described their overall health as good (28.3%), very good (44.9%), or excellent (16.6%), while 9% described their health as fair and 1.2% as poor. Additionally, the majority (94.2%) of mothers described their child's health as good (8.5%), very good (29%) or excellent (56.7%), while 4% endorsed fair, and 0.9% endorsed poor. Regarding sleep quality over the past month, 27.9% described their sleep as very bad or fairly bad, 36.8% as okay, and 35.3% as fairly good or very good. The majority of respondents (76.2%) reported incorporating fresh fruits and vegetables into their diets more than three times per week. Participants reported engaging in 30 minutes of moderate to vigorous exercise 3.57 days per week on average, with 22% of the sample reporting exercise less than once per week, 26.6% reporting one or two days per week, 33.6% reporting three or four days per week, 13.4% reporting five or six days per week, and 4.3% reporting exercising 7 days per week. The majority (83.2%) of mothers endorsed feeling either moderately (27%), somewhat (32.4%), or completely (23.8%) satisfied with their perceived level of social support. Additionally, social support had a significant negative correlation ($r = -.39$, $p < .001$) with wellbeing (OQ-45), indicating that those

with a higher social support satisfaction reported less overall distress as measured by the OQ-45.

Participants were asked about their experience of significant or adverse events in the past 12 months within health, interpersonal, occupational, financial, and legal domains. In total, 28.2% of the sample reported pregnancy or giving birth, and 19.9% reported that they, their partner, or a close relative experienced a serious illness, injury or assault. Additionally, 29.7% experienced the death of a parent, partner, child, or close relative or friend, and 4.7% reported living with someone with an alcohol or drug issue. When asked about employment and finances, 14.5% described feeling like they would soon lose their job, 10.9% reported losing their job, 15.5% reported unsuccessfully seeking employment for at least one month, and 18.6% reported experiencing a major financial crisis. Refer to Table 1 for more information on health and wellbeing related variables.

**Mental health.** One third (32.1%) of the sample self-reported a previous mental health diagnosis, which included comorbid anxiety disorders and depression (33.9%), depression (25.8%), anxiety (23.5%), and other (16.8%). Refer to Table 1 for more information on self-reports of mental health diagnoses. The mean score on the OQ-45 was 54.46, with 30.2% of mothers obtaining a score above the clinical cut-off (total score of 63) for psychological distress. There was a strong, negative correlation between mother's perception of their personal health and their total OQ-45 scores ($r_s(1080) = -.450$, $p < .001$), suggesting that mothers with a negative perception of their health indicated higher psychological distress on the OQ-45.

## Parenting

The mean score for maternal satisfaction on the parenting sense of competence measure was 36.60 and the mean score for efficacy was 36.24. Scores ranging from 36.38 to 41.00 for maternal satisfaction and 21.53 to 30.50 for maternal efficacy have been suggested as normative in a study validating this scale within a Canadian community sample of mothers with children aged 5–12 (Ohan et al., 2000). On a scale assessing mothers' family life satisfaction (KFLS), the mean score was 5.73, indicating low levels of satisfaction with familial relationships within this sample. Similarly, the mean score for child-parent closeness was 16.00, indicating relatively low levels of closeness, while the mean score for child-parent conflict was 31.64, indicating high-average levels of conflict overall. Mothers' mean score for conflict with current partners was 25.32 while the mean score for support was 11.88, suggesting mothers perceived low levels of supportive communication and high levels of conflicting conversation with co-parents. The mean and range scores for the family scales can be located in Table 2.

## Service need,utilization, and satisfaction

When asked about service utilization for their children, mothers most frequently reported accessing their family doctor (84.2%), dental services (56.2%), and the hospital emergency department (45.8%). The services mothers' most frequently reported needing but being unable to access for their children were youth health information from phone or internet services (38.5%), psychiatric or behavioural services (27.3%), and other medical specialists (26.2%). In regard to service utilization for themselves, mothers most frequently reported using parenting information from phone or internet services (18%), counselling (16.1%), and parenting support groups (11.6%). The services mothers' most frequently reported needing but being unable to access were adult mental health services (35.3%), other counselling services (28.7%), and relationship counselling services (21.3%). Refer to Table 3 for more information on service utilization and needs. Satisfaction with the availability of services was rated highest for nursing

**Table 2. Descriptive results for the psychometric parenting and well-being measures of the sample who completed the Motherhood survey.**

| Scale | n | Mean (SD) | Range |
|---|---|---|---|
| Outcome Questionnaire | 1013 | 54.462 (22.920) | 5–144 |
| Missing Data | 69 | | |
| Kansas Family Life Satisfaction Scale | 1051 | 5.726 (1.059) | 1–7 |
| Missing Data | 31 | | |
| Parenting Sense of Competence Scale (PSOC) | 1063 | 72.834 (11.153) | 36–102 |
| PSOC Satisfaction | | 36.598 (7.394) | 9–54 |
| PSOC Efficacy | | 36.235 (5.508) | 20–48 |
| Missing Data | 19 | | |
| Quality of Co-Parenting Communication Scale Current Partner | 868 | 34.319 (4.481) | 10–47 |
| Closeness Subscale | 13 | 9.005 (2.766) | 4–20 |
| Conflict Subscale | | 25.315 (4.590) | 4–20 |
| Missing Data | | | |
| Quality of Co-Parenting Communication Scale Former Partner | 142 | 30. 880 (4.756) | 10–50 |
| Closeness Subscale | 18 | 11.880 (4.083) | 4–20 |
| Conflict Subscale | | 19.000 (5.019) | 6–30 |
| Missing Data | | | |
| Child Parent Relationship Scale (Short Form) Biological | 1008 | 47.682 (7.423) | 23–75 |
| Closeness Subscale | 3 | 16.040 (6.492) | 3–40 |
| Conflict Subscale | | 31.640 (3.620) | 16–35 |
| Missing Data | | | |
| Child Parent Relationship Scale (Short Form) Step-Child | 53 | 45.453 (8.900) | 31–75 |
| Closeness Subscale | | 18.029 (8.505) | 8–40 |
| Conflict Subscale | | 27.434 (6.090) | 11–35 |
| Child Parent Relationship Scale (Short Form) Adopted | 17 | 51. 177 (7.552) | 37–64 |
| Closeness Subscale | | 20.412 (8.071) | 9–38 |
| Conflict Subscale | | 30.765 (4.549) | 20–35 |
| Weekly Produce Consumption | 1082 | 2.70 (.597) | 0–3 |
| Weekly Exercise Frequency | 1082 | 3.57 (1.976) | 0–7 |
| Sleep Quality (past month) | 1082 | 3.10 (1.035) | 1–5 |
| Personal Health Rating | 1082 | 3.67 (.898) | 1–5 |
| Child Health Rating | 1082 | 4.37 (.87) | 1–5 |
| Social Support Satisfaction | 1082 | 3.58 (1.119) | 1–5 |

services (M = 3.86, SD = .95), followed by physician services (M = 3.81, SD = 1.22), other health services (M = 3.33, SD = 1.13), support groups (M = 3.12, SD = .99), and mental health services (M = 2.74, SD = 1.10).

## Predicting Mothers' adult service initiation

The results of the binomial logistic regression analysis of mothers' initiation of adult services are reported in Table 4. In this sample, 50.6% (n = 547) of mothers endorsed accessing at least one service in the previous 12 months. STEP1 contained sociodemographic covariates, was statistically significant $X^2$ (9) = 87.35, $p$ < .001, and explained 10.6% of the variance. Four of the nine predisposing factors were statistically significant predictors: maternal age, maternal education attainment, perceived availability of services, and marital status. As mothers' age decreased the likelihood of accessing at least one service increased ($p$ = .016) while increasing maternal educational attainment was associated with an increased likelihood ($p$ < .001) of accessing at least one service. When services were perceived to be available, there was 3.88 higher odds of mothers accessing at least one service ($p$ < .001). While statistically significant in the regression ($p$ = .03), a one-way ANOVA follow up test revealed that adult service utilization did not differ across marital status ($p$ = .22).

**Table 3. Descriptive statistics of services accessed and need for the sample.**

| Variables | N | % | Missing Data |
|---|---|---|---|
| Services Accessed for Child(ren) (past 12 months) | *1075* | | 7 |
| Youth health info from phone/internet | 321 | 29.9 | |
| Hospital emergency ward | 492 | 45.8 | |
| Hospital outpatient clinic | 356 | 33.1 | |
| GP or family doctor | 905 | 84.2 | |
| Disability services | 36 | 3.3 | |
| Speech therapy | 96 | 8.9 | |
| Dental services | 604 | 56.2 | |
| Pediatrician | 274 | 25.5 | |
| Guidance Counsellor | 103 | 9.6 | |
| Other psychiatric/behavioural service | 91 | 8.5 | |
| Other medical specialist | 254 | 23.6 | |
| None of the above | 42 | 3.9 | |
| Services Needed but not Available for Child(ren) | *187* | | 895 |
| Youth health info from phone/internet | 72 | 38.5 | |
| Hospital emergency ward | 8 | 4.3 | |
| Hospital outpatient clinic | 5 | 2.7 | |
| GP or family doctor | 29 | 15.5 | |
| Disability services | 11 | 5.9 | |
| Speech therapy | 22 | 11.8 | |
| Dental services | 23 | 12.3 | |
| Pediatrician | 28 | 15.0 | |
| Guidance Counsellor | 4 | 2.1 | |
| Other psychiatric/behavioural service | 51 | 27.3 | |
| Other medical specialist | 49 | 26.2 | |
| None of the above | 0 | 0 | |
| Services Accessed for Family (past 12 months) | *1071* | | 11 |
| Parenting education courses | 107 | 10.0 | |
| Relationship education courses | 7 | 0.7 | |
| Relationship counselling | 44 | 4.1 | |
| Other counselling services | 172 | 16.1 | |
| Parent support group | 124 | 11.6 | |
| Parent information from phone or internet | 193 | 18.0 | |
| Drug or alcohol services | 16 | 1.5 | |
| Problem gambling services | 1 | 0.1 | |
| Adult mental health services | 121 | 11.3 | |
| Migrant or ethnic resource services | 0 | 0.0 | |
| Housing services | 23 | 2.1 | |
| Disability services | 24 | 2.2 | |
| Financial management services | 62 | 5.8 | |
| Charities | 31 | 2.9 | |
| Emergency relief services | 2 | 0.2 | |
| Church or religious group | 59 | 5.5 | |
| Other family support services | 53 | 4.9 | |
| None of the above | 542 | 50.6 | |
| Services Needed but not Available for Family | *150* | | |

*(Continued)*

**Table 3.** (Continued)

| Variables | N | % | Missing Data |
|---|---|---|---|
| Parenting education courses | 23 | 15.3 | |
| Relationship education courses | 9 | 6.0 | |
| Relationship counselling | 32 | 21.3 | |
| Other counselling services | 43 | 28.7 | |
| Parent support group | 15 | 10.0 | |
| Parent information from phone or internet | 3 | 2.0 | |
| Drug or alcohol services | 7 | 4.7 | |
| Problem gambling services | 2 | 1.3 | |
| Adult mental health services | 53 | 35.3 | |
| Migrant or ethnic resource services | 1 | 0.7 | |
| Housing services | 15 | 10.0 | |
| Disability services | 11 | 7.3 | |
| Financial management services | 15 | 10.0 | |
| Charities | 2 | 1.3 | |
| Emergency relief services | 3 | 2.0 | |
| Church or religious group | 2 | 1.3 | |
| Other family support services | 12 | 8.0 | |
| None of the above | 0 | 0 | |

**Table 4. Binomial Logistic Regression Analysis of Mothers' Adult Service Initiation.**

| Independent Variables | B | SE | Wald | df | p | Odds Ratio | 95% CI for Odds Ratio | |
|---|---|---|---|---|---|---|---|---|
| | | | | | | | LL | UP |
| Step 1 | | | | | | | | |
| Population Centre | -.130 | .123 | 1.122 | 1 | .289 | .878 | .691 | 1.117 |
| Geographical location (urban/rural) | .176 | .308 | .326 | 1 | .568 | 1.192 | .652 | 2.181 |
| Maternal age | -.022 | .009 | 5.810 | 1 | .016 | .979 | .961 | .996 |
| Household income | -.005 | .034 | .020 | 1 | .889 | .995 | .932 | 1.063 |
| Maternal educational attainment | .197 | .040 | 23.937 | 1 | .000 | 1.217 | 1.125 | 1.317 |
| Maternal occupation status | -.258 | .146 | 3.140 | 1 | .076 | .772 | .580 | 1.028 |
| Maternal marital status | .184 | .085 | 4.688 | 1 | .033 | 1.203 | 1.018 | 1.421 |
| Type of child (biological, step, adopted) | -.155 | .223 | .484 | 1 | .487 | .856 | .553 | 1.326 |
| Availability of adult services | 1.357 | .213 | 40.613 | 1 | .000 | 3.884 | 2.559 | 5.895 |
| Step 2 | | | | | | | | |
| Biological CPRS Closeness | -.059 | .017 | 12.124 | 1 | .000 | .942 | .912 | .974 |
| Current Partner QCPC Conflict | .087 | .025 | 12.117 | 1 | .000 | 1.090 | 1.039 | 1.145 |
| Current Partner QCPC Support | .037 | .016 | 5.568 | 1 | .018 | 1.038 | 1.006 | 1.071 |
| Satisfaction with total health service availability | .203 | .094 | 4.642 | 1 | .031 | 1.225 | 1.019 | 1.474 |
| Total KFLS | -.167 | .075 | 4.976 | 1 | .026 | .846 | .731 | .980 |
| Step 3 | | | | | | | | |
| Maternal mental health diagnosis | .815 | .147 | 30.714 | 1 | .000 | 2.258 | 1.693 | 3.012 |
| OQ total score | .009 | .004 | 4.830 | 1 | .028 | 1.009 | 1.001 | 1.018 |
| Rating of child's health | -.171 | .083 | 4.214 | 1 | .040 | .843 | .715 | .992 |
| Step 4 | | | | | | | | |
| Total child service utilization | .192 | .043 | 20.296 | 1 | .000 | 1.212 | 1.115 | 1.317 |
| Weekly fresh produce | .274 | .129 | 4.513 | 1 | .034 | 1.316 | 1.021 | 1.695 |

*Note*: B = Unstandardized regression coefficient; CPRS = Child-Parent Relationship Scale; KFLS = Kansas Family Life Satisfaction Scale; OQ = Outcome Questionnaire; QCPC = Quality of Co-Parental Communication Scale

Five of the eleven indicators of enabling resources entered in STEP2 were significant predictors and explained an additional 4.7% of unique variance when included in the model. Increases in close ($p$ = .018) and conflictual ($p$ < .001) communication with current partners were associated with increased likelihood of accessing any services. Decreases in closeness within parent-child relationships ($p$ < .001), and in family life satisfaction ($p$ = .026) were associated with increased likelihood of accessing services. Finally, increased satisfaction with availability of services was associated with increased likelihood of accessing services ($p$ = .023).

Three of the four indicators of health need entered in STEP3 were significant predictors, and explained an additional 4.4% of unique variance when included in the model. Decreases in mothers' ratings of their children's overall health was associated with increased likelihood of mothers accessing adult services ($p$ = .040). Additionally, increases in mental health symptomatology were associated with increased likelihood of accessing services ($p$ = .028) and mothers who self-reported having a mental health diagnosis had 2.26 higher odds of accessing services than mothers who self-reported not having a mental health diagnosis ($p$ < .001).

An increase in child services accessed in the previous 12 months ($p$ < .001), and weekly produce consumption ($p$ = .034) were entered into the final step, and explained 2.8% of unique variance. The logistic regression model was significant, $X^2$ (19) = 194.72, $p$ < .001, and the final model explained a total of 22.5% of variance in maternal use of adult healthcare services in the previous 12 months with 66.6% of cases correctly classified. Sensitivity was 64.3%, and specificity 68.8%.

## Predicting Mothers' child service initiation

The results of the binomial logistic regression analysis of mothers' initiation of services for their children are reported in Table 5. In this sample, 95.7% (n = 1036) of mothers endorsed

**Table 5. Binomial Logistic Regression Analysis of Mothers' Child Service Initiation.**

| Independent Variables | *B* | SE | Wald | *df* | *p* | Odds Ratio | 95% CI for Odds Ratio | |
|---|---|---|---|---|---|---|---|---|
| | | | | | | | LL | UP |
| Step 1 | | | | | | | | |
| Population Centre | -.696 | .314 | 4.918 | 1 | .027 | .499 | .270 | .922 |
| Geographical location (urban or rural) | 1.959 | .817 | 5.753 | 1 | .016 | 7.093 | 1.431 | 35.159 |
| Maternal age | -.093 | .020 | 22.010 | 1 | .000 | .911 | .876 | .947 |
| Household income | .141 | .085 | 2.782 | 1 | .095 | 1.152 | .976 | 1.360 |
| Maternal educational attainment | .014 | .103 | .020 | 1 | .888 | 1.015 | .829 | 1.241 |
| Maternal occupation status | .574 | .386 | 2.215 | 1 | .137 | 1.775 | .834 | 3.780 |
| Maternal marital status | .068 | .173 | .156 | 1 | .693 | 1.071 | .763 | 1.502 |
| Type of child (biological, step, or adopted) | -1.591 | .409 | 15.140 | 1 | .000 | .204 | .091 | .454 |
| Step 2 | | | | | | | | |
| Maternal frequency of contact with social support | -.693 | .306 | 5.108 | 1 | .024 | .500 | .274 | .912 |
| Step 3 | | | | | | | | |
| Maternal mental health diagnosis | 1.547 | .539 | 8.232 | 1 | .004 | 4.698 | 1.633 | 13.516 |
| Rating of child's health | -.704 | .266 | 6.984 | 1 | .008 | .494 | .293 | .834 |
| Step 4 | | | | | | | | |
| Maternal weekly exercise frequency | -.211 | .089 | 5.605 | 1 | .018 | .810 | .810 | .681 |

*Note*: B = Unstandardized regression coefficient

accessing at least one service for their child in the previous 12 months. STEP1 of the binomial logistic regression contained sociodemographic covariates, was statistically significant $X^2$ (8) = 98.02, $p < .001$, and explained 29.4% of the variance. Four of the eight environmental and predisposing factors were significant predictors: population centre, geographical location, maternal age, and type of child (refer to Table 5). As the size of the community mothers were living in increased so did the likelihood of accessing child services ($p = .027$), and mothers living in urban locations had 7.09 higher odds of accessing at least one child service compared to mothers living in rural areas ($p = .016$). Increase in mother's age was associated with decreased likelihood of accessing at least one child service ($p < .001$). Finally, mothers with only biological children were less likely to access child services.

One of the eleven indicators of enabling resources entered in STEP2 was included in the model. Maternal frequency of contact with social supports explained an additional 1.6% of the unique variance. Decreases in mothers' frequency of contact with social supports was associated with increased likelihood of mothers accessing at least one child service ($p = .024$).

Two of the four indicators of need characteristics entered in STEP3 were included in the model and explained an additional 5.5% of the unique variance. Decreases in mothers' ratings of their children's overall health was associated with increased likelihood of mothers accessing at least one child service ($p = .008$), and mothers who self-reported a mental health diagnosis had 4.70-fold higher odds of accessing at least one service for their children relative to mothers who did not ($p = .004$).

In STEP4, one of the four indicators of health behaviours was included in the model and explained an additional 1.6% of unique variance. As the frequency of maternal weekly exercise decreased, the likelihood of accessing at least one service increased ($p = .018$). The final logistic regression model was statistically significant $X^2$ (12) = 128.66, $p < .001$, and explained 38.3% of the variance in maternal access of at least one child service in the previous 12 months with 96.3% of cases correctly classified. Sensitivity was 99.4%, and specificity 28.3%.

### Predicting Mothers' continuous utilization of adult services

The results of the linear regression analysis of mothers' continuous use of a range of adult services are reported in Table 6. Only mothers who reported accessing at least one service (n = 548) were included. STEP1 included covariates and was statistically significant, $F(9, 536)$ = 9.47, $p < .001$, accounting for 13.7% of the unique variance. Three of the nine environmental and predisposing predictor variables were statistically significant. As mothers' age ($\beta$ = -0.085, $p = .05$) and household income ($\beta$ = -.178, $p = .002$) decreased, their likelihood of accessing a range of services increased, while mothers who felt that additional services were needed reported accessing fewer services than mothers who reported that required services were available ($\beta$ = .263, $p < .001$).

In STEP2, only one of the eleven variables entered was statistically significant and included in the model, F (1, 535) = 11.38, $p = .001$, accounting for an additional 1.8% of unique variance. Increases in family life satisfaction were associated with decreases in continuous adult service utilization ($\beta$ = -.159, $p < .001$)

Two of the four health need variables entered into STEP3 of the regression were statistically significant and included in the overall model, F (2, 533) = 9.20, $p < .001$, accounting for an additional 2.8% of unique variance. Decreases in mother's overall ratings of their own health ($\beta$ = -.146, $p = .001$) and their child's health ($\beta$ = -.106, $p = .014$) were associated with an increase in the number of self-reported adult services utilized.

In STEP4, of the four health practice variables included in the regression, only total services accessed for children had an independent effect, F (1, 532) = 23.96, $p < .001$ and

**Table 6. Binomial Linear Regression Analysis of Mothers' Continuous Use of a Range of Adult Services.**

| Independent Variables | Model 1 | | Model 2 | | Model 3 | | Model 4 | | Model 5 | |
|---|---|---|---|---|---|---|---|---|---|---|
| | $\beta$ | t | $\beta$ | t | $\beta$ | t | $\beta$ | t | $\beta$ | t |
| Population Centre | -.180 | -1.862 | -.160 | -1.669 | -.161 | -1.697 | -.160 | -1.698 | -.126 | -1.358 |
| Geographical location (urban or rural) | .136 | 1.413 | .123 | 1.297 | .127 | 1.346 | .126 | 1.340 | .089 | .967 |
| Maternal age | -.085 | -1.9098 | -.113 | -2.519** | -.097 | -2.179* | -.104 | -2.335* | -.096 | -2.204* |
| Household income | -.178 | -3.169*** | -.167 | -3.005*** | -.135 | -2.415* | -.145 | -2.605** | -.152 | -2.787** |
| Maternal educational attainment | -.021 | -.446 | -.024 | -.508 | -.007 | -.156 | .004 | .089 | -.013 | -.292 |
| Maternal occupation status | -.015 | -.360 | -.026 | -.613 | -.028 | -.659 | -.031 | -.745 | -.041 | -1.003 |
| Maternal marital status | .013 | .274 | -.035 | -.704 | -.015 | -.297 | -.017 | -.345 | -.007 | -.151 |
| Type of child (biological, step, or adopted) | .024 | .568 | .020 | .476 | .024 | .572 | .009 | .208 | .018 | .438 |
| Availability of adult services | .263 | 6.230*** | .232 | 5.450*** | .223 | 5.271*** | .216 | 5.120*** | .194 | 4.672*** |
| KFLS total score | | | -.159 | -3.511*** | -.126 | -2.736** | -.116 | -2.538** | -.128 | -2.835*** |
| Personal health rating | | | | | -.146 | -3.397*** | -.119 | -2.688** | -.104 | -2.389** |
| Child health rating | | | | | | | -.106 | -2.477** | -.026 | -.574* |
| Total paediatric service utilization | | | | | | | | | .206 | 4.798*** |

*Note*: $\beta$ = Standardized regression coefficient; KFLS = Kansas Family Life Satisfaction Scale

\* $p < .05$.

\*\* $p < .01$.

\*\*\* $p < .005$

accounted for an additional 3.5% of the unique variance. As service use for children increased, the likelihood of mothers utilizing a range of adult services increased as well ($\beta$ = .206, $p < .001$). The final model accounted for 21.8% of the variance in maternal use of a range of adult health services.

## Predicting Mothers' continuous utilization of child services

The results of the linear regression analysis of mothers' continuous use of a range of child services are reported in Table 7. Only mothers who reported accessing at least one child service (n = 1,036) were included. STEP1 included covariates and was not statistically significant, F (8, 1008) = 0.53, $p = .836$.

Two of the eleven variables entered in STEP2, were included in the model, F (3, 1005) = 8.16, $p < .001$, and accounted for 2.4% of unique variance. Increases in mother's sense of parenting satisfaction ($\beta$ = -.106, $p = .001$), and mother's family life satisfaction ($\beta$ = -.081, $p = .031$) were associated with decreases in continuous use of child services, while increases in supportive communication with current partners was associated with increases in utilization ($\beta$ = .097, $p = .004$).

Two of the four health need variables included in STEP3 were included in the model, F(2, 1003) = 87.24, $p < .001$, and accounted for 14.4% of unique variance. Mothers who reported poorer ratings of their child's health ($\beta$ = -.362, $p < .001$), or having a mental health diagnosis ($\beta$ = .162, p < .001) were more likely to continue using a range of child services.

In STEP4, only one of the three health practice variables entered into the regression was statistically significant and included in the overall model, F (1, 1002) = 8.55, $p = .004$, accounting for an additional 0.7% of the unique variance. As maternal sleep quality improved, the likelihood of mothers continued use of a range of child services decreased ($\beta$ = -.089, $p = .003$). The final model accounted for 17.9% of the variance in maternal continued use of child services.

**Table 7. Linear Regression Analysis of Mothers' Continuous Use of a Range of Child Services.**

| Independent Variables | Model 1 | | Model 2 | | Model 3 | | Model 4 | | Model 5 | | Model 6 | | Model 7 | |
|---|---|---|---|---|---|---|---|---|---|---|---|---|---|---|
| | β | t | β | t | β | t | β | t | β | t | β | t | β | t |
| Population Centre | -.013 | -.171 | -.010 | -.138 | -.019 | -.248 | -.028 | -.375 | -.044 | -.628 | -.033 | -.471 | -.036 | -.519 |
| Geographical location (urban or rural) | .022 | .292 | .022 | .299 | .041 | .547 | .053 | .707 | .059 | .845 | .063 | .910 | .065 | .943 |
| Maternal age | -.015 | -.435 | -.009 | -.248 | -.001 | -.029 | -.017 | -.479 | -.048 | -1.464 | -.046 | -1.426 | -.037 | -1.132 |
| Household income | -.012 | -.286 | .002 | .043 | -.005 | -.116 | .000 | -.010 | .010 | .257 | .013 | .341 | .018 | .473 |
| Maternal educational attainment | .046 | 1.264 | .049 | 1.361 | .046 | 1.261 | .042 | 1.159 | .075 | 2.215* | .088 | 2.623** | .090 | 2.682 |
| Maternal occupation status | .021 | .622 | .030 | .899 | .030 | .915 | .028 | .839 | .015 | .472 | .017 | .559 | .025 | .818 |
| Maternal marital status | -.040 | -1.082 | -.031 | -.845 | -.012 | -.332 | -.035 | -.924 | -.029 | -.810 | -.035 | -.979 | -.027 | -.765 |
| Type of child (biological, step, or adopted) | .012 | .352 | .011 | .337 | .017 | .528 | .020 | .600 | -.019 | -.617 | -.012 | -.392 | -.006 | -.191 |
| PSCS satisfaction | | | -.106 | -3.325*** | -.124 | -3.828*** | -.101 | -2.969*** | -.054 | -1.679*** | -.032 | -1.008*** | -.024 | -.762*** |
| QCPC Support | | | | | .097 | 2.897*** | .114 | 3.334*** | .112 | 3.479*** | .100 | 3.143*** | .105 | 3.305*** |
| KFLS total score | | | | | | | -.081 | -2.164* | -.027 | -.778* | -.024 | -.691* | -.012 | -.336* |
| Child health rating | | | | | | | | | -.362 | -11.851*** | -.350 | -11.585*** | -.342 | -11.313*** |
| Mental health diagnosis | | | | | | | | | | | .162 | 5.491*** | .154 | 5.208*** |
| Maternal sleep quality | | | | | | | | | | | | | -.089 | -2.944*** |

*Note*: β = Standardized regression coefficient; KFLS = Kansas Family Life Satisfaction Scale; PSCS = Parental Sense of Competence Scale; QCPC = Quality of Co-Parental Communication Scale

* $p < .05$.

** $p < .01$.

*** $p < .005$

## Discussion

This was an explorative provincial cross-sectional survey study of maternal HSU among a sample of 1,082 mothers. This sample is the largest known survey of demographic information and service use of mothers in Canada. The survey's specificity to NL allows for better understanding of this geographically and culturally unique group. The demographic profile, including women's overall health, mental health, and service use and need will be outlined below. Mother's patterns of HSU for themselves and their children will then be summarized within Andersen's Behavioural Health Model which supports a parsimonious understanding of multiple determinants of HSU at an individual and contextual level.

### Demographic profile of health and service need

Most participants experience strong social support in their communities, engage in positive health behaviours (e.g., vegetable consumption, exercise, sleep), and view themselves and their children as relatively healthy. In comparison to research across Canada indicating 1 in 5 people experience a mental health challenge [86, 93] approximately one third of the sample reported experiencing mental health challenges and indicated that mental health services were the most inaccessible. This is concerning given that duration of untreated mental illness is associated with poor outcomes among those who experience serious (e.g., psychosis, bipolar disorder) and common mental illnesses (e.g., depression, anxiety) [94–96]. Additionally, untreated mental illness is associated with overuse of primary care services, loss of wages and workplace productivity, and poorer physical health [97–99]. Previous research suggests that a high percentage of individuals report initially looking to their primary care physician for mental

health support [99, 100]. Given the prevalence of mental health needs of mothers in the community and dissatisfaction with the availability of mental health services in the community, coupled with the higher sense of satisfaction with availability of GPs, healthcare policies directed towards integrating mental healthcare into primary care settings may facilitate improved access to services.

## Predicting Mothers' access to adult services

While previous research has illuminated the relationship between psychosocial factors and service utilization, there has been a scarcity of research on which of these established factors apply to the population of mothers. The results of this study suggest that environmental, predisposing, enabling, need, and personal health practice factors independently predict Canadian mothers' initial and continued HSU, supporting Andersen's proposition that each component of the Behavioural Health Model contributes to predicting service utilization [12]. The different constellation of variables observed to predict mothers initiated and continued access to services is consistent with previous literature that has reported common and unique factors that impact both dimensions of access to care [12, 18, 101]. The greater number of variables included in the model predicting mothers' initial entry into services compared to continued access is congruent with the contention that initial establishment of contact with the healthcare system is impacted by a wider spectrum of factors beyond need as compared to ongoing service utilization [102, 103].

The results of this study suggest that access to initial and on-going care remains inequitable for mothers despite Canada's healthcare system being publicly funded, with the most predictive factor of access to care being environmental and predisposing variables rather than need variables. The relevance of maternal education attainment, household income, and perceived availability of services on HSU is consistent with previous observations of the widening socio-economic inequalities in access to care for women, despite improvements in Canadian's overall health and increases in per capita health spending over the past four decades. [65, 104]. Consistent with previous research suggesting that motherhood may motivate help-seeking behaviour, increases in mothers' mental health symptomatology was associated with increased likelihood of initiating HSU and decreases in overall health were associated with continued HSU [97–99, 105–107].

Utilization of child services was associated with utilization of adult services which supports the hypothesis that availing of child services supports maternal health through screening and referrals for maternal mental health, behavioural health, and social needs [108]. Indeed, the Canadian Paediatric Society recommended that paediatric clinicians support positive parenting through the provision of care, and referrals to specialist services and supports where appropriate [109].

## Predicting Mothers' access to child services

Few studies have considered the influence of maternal context on children's HSU despite the hegemonic expectations on mothers to take responsibility for attending to the health of their children [29, 33, 52, 53, 110], and research demonstrating that mothers are typically the predominate caregiver of children and significantly impact their children's health [29, 33, 53, 110, 111]. Previous scoping and systematic reviews on research examining access to child services has identified significant limitations in the measurement of family variables, a paucity of studies examining community populations with a range of mental health concerns, and limited literature examining the impact of environmental factors (e.g., region), predisposing characteristics, and need factors [32, 35, 104, 112]. The results of the present study

demonstrate the relevance of these factors, as well as enabling and personal health practice factors for understanding child HSU. Each of these categories of maternal variables collectively predicted mothers initial use of child services while enabling, need, and personal health practices predicted continued use of services. Specifically, region of residence, maternal age, maternal social support, maternal and child health status, and maternal personal health practices were observed to impact children's initial access to care. While these results suggest ongoing environmental and social disparities in initial access to child HSU, the absence of environmental and predisposing variables from the continued access model suggests that ongoing access to child services may be more equitable than access to adult services. Specifically, once mothers have accessed the healthcare system maternal and child health status as well as maternal personal health practices serve as reinforcing factors for child HSU, while maternal role satisfaction, maternal relationship quality, and co-parental communication become relevant determinants in the decision to continue pursuing access to care.

The observation that rurality of the community was a predictor of initiation of child services in NL is consistent with previous research demonstrating that children residing in rural areas have greater difficulty accessing healthcare than their urban counterparts and have higher unmet needs for medical, dental, and mental health [113–115]. Documented system-related barriers to accessing child health services for those living outside of urban centers include shortage of providers, protracted wait times, lack of coordination of clinical services, insufficient funding, a need to train more primary care physicians working in rural areas, and transportation [35, 113, 115–119]. These barriers are often compounded by socioeconomic and attitudinal barriers, exacerbating difficulties accessing care in rural areas [115, 117]. Training a greater volume of care providers, providing additional support and education for rural providers, delivering services through integrated health teams, and the adoption of telemedicine are some evidence-based approaches for supporting access to care in rural areas [116, 117, 120, 121].

## Clinical implications

The relationship between maternal wellbeing, child outcomes, and family functioning highlights the importance of targeting maternal access to services to meet health needs of mothers and their children. Mothers experiencing mental health challenges in this study were more likely to access services for themselves and their children relative to those who did not report clinically-relevant levels of psychological distress. This suggests that Canadian mothers residing in the community have positive attitudes towards help-seeking for mental health needs and are willing to utilize resources made available to them. That said, healthcare professionals should continue to undergo training to recognize entrenched biases towards mother blaming and maintain efforts not to communicate unwarranted value judgements to mothers seeking care given prevailing stigma towards mental illness and research demonstrating the experience of mother-blame when accessing health resources [28, 29, 59, 121]. This is especially important given the observation that satisfaction with healthcare providers impacts mothers access to services [8, 12, 122].

Further, the decision to use healthcare services is a complex result of an interplay of multiple static and dynamic factors. Results of this study suggest that family relationships play a significant role in mothers' decision to access healthcare given that high levels of social support, quality of familial relationships, role satisfaction, and style of communication significantly reduced mothers' need for services. This observation suggests that parenting programs, family therapy, and family services that focus on teaching effective communication skills (e.g., active listening, empathetic responding), strengthening emotional bonds, and supporting family

cohesion may reduce mothers' overall need for services. Another significant factor in mothers HSU for themselves and their children was mothers' mental health and children's health status. While previous research has demonstrated the impacts of maternal wellbeing on child health, there has been less examination on the impacts of children's health on maternal health and even less literature on how this relationship impacts access to services. By uniting children and maternal health into one model for HSU, the results of this study suggest that children and mothers' health and service use is highly interdependent, and that policy efforts to improve access for one may have benefits for the other. Further, the reciprocal relationship observed between mothers' and children's health and service use suggest the need for health policies that target families as a collective in addition to individuals.

Policy directives aimed at addressing structural barriers to care (e.g., availability, maternal education, financial resources) are also needed to improve equitable access to care for mothers and children. In the present study mothers reported that family physicians and emergency departments were the most frequently utilized service for youth. Youth mental health information, psychiatric and behavioural services, and other medical specialist services were the most inaccessible services. This is consistent with other research suggesting that 15% to 40% of children with mental health challenges are seen by GPs, who often feel ill equipped and inadequately trained to address these needs [115, 119]. Further, mothers with less education, lower household income, and living in less populated and more rural areas were less likely to access services for themselves and their children. These observations are consistent with previous research indicating a pro-rich bias for access to specialist care in Canada and the limited availability of mental healthcare in rural Canada [18, 65, 123, 124]. Collectively, these observations suggest that socioeconomic barriers to services remains an issue despite the aims of Canada's publicly funded healthcare system. Inequitable access to care may contribute to the documented widening of socioeconomic disparities in Canadians health, especially among women [125].

### Limitations and future directions

There are several limitations of this study. First, implications of the homogenous composition of the sample are worth noting. While significant efforts to recruit a diverse sample of mothers residing within NL were made, the final sample under-represented mothers from diverse ethnic and socio-economic backgrounds, mothers from Labrador, and mothers of adopted and/ or stepchildren. While NL is a predominately Caucasian population with the majority of children residing with biological parents [93], this overall lack of diversity limits the generalizability of these findings to some sub-groups of the population of mothers. This sample is, however, one of the largest known collections of demographic, health, and service use information of mothers in Canada, and its specificity to mothers in NL was intended to help better understand this culturally and geographically unique group. While variables assessed were selected based on broad literature of HSU, women, and families, it is possible that the observations of this study may not generalize to mothers living in other provinces given the unique characteristics of NL culture. A replication study within other provinces in Atlantic Canada may be beneficial to determine if similar relationships are observed and test the interpretations made by this study of the relationship between these factors and HSU.

The cross-sectional design of this study means that all results are correlational, which precludes the ability to infer directionality or causality. Further, the present study was a secondary data analysis from an online survey designed for another project, which limited variables available for analysis. While the dataset was comprehensive, inclusion of additional socio-demographic variables, personal health attitudes, and perceptions of healthcare providers would

have supported more in-depth understanding of the goals of this project. Additionally, in an effort to increase participation and address maternal experience specifically, in the present survey mothers were asked to report only on the child with the next birthday rather than all children and many questions related to the family more broadly were not included. Lastly, beliefs related to parenting and personal attitudes regarding healthcare utilization, such as beliefs in the value of preventative care and mental healthcare, stigma, and perceptions of mother blame by providers could have been beneficial for further illuminating how personal attitudes and societal norms of intensive mothering manifest in healthcare and impact service utilization.

There is a dearth of research on HSU among normative samples of mothers. Women are usually the focus of research related to motherhood when they have been identified as 'at risk' (e.g., teen mothers, disadvantaged single mothers, immigrant mothers, mothers who present with severe mental illness), and findings are often framed in a deficit perspective which leaves broad arrays of maternal experiences understudied. Research on HSU among mothers follows this pattern (e.g., [25, 126–131]. As such, it is crucial to conduct research on a normative sample of community-dwelling mothers to understand general motherhood experiences (Arendell, 2000). Further, observations from this study suggest that implementation of integrative, and coordinated care for families may be an effective approach for improving access to care. Studies completed in the UK have provided empirical evidence for the benefits of integrated care on meeting healthcare needs [132–134]. Program development of integrated care models and program evaluation research of this model in Canada are needed to assess how this model can be implemented in Canada and the efficacy of it for improving access to care. Finally, more research is needed that utilizes a feminist lens and takes family context into consideration to understand mothers HSU. The importance of family relationships and the interplay between child and mother health status and service use observed demonstrates the utility of understanding these variables to support access to care.

## Conclusion

This study emphasized the need for a gendered understanding of health and access to care, particularly for maternal health. Access to services is a critical determinant of individual and population health outcomes and understanding HSU is important for planning of resource allocation in the community and addressing barriers to care. This study has filled an important gap in our knowledge and illuminated factors that impact access to services among Canadian mothers and their children. The results highlight that there was a need for improved access to mental health services despite mothers in NL demonstrating strong social support and perceiving themselves and their children as healthy. The study also underscores the significance of integrated healthcare services for families and the different factors influencing mothers' initiation and continuation of healthcare services. Overall, further investigation is required to understand how prevailing gender ideologies impact mothers, the factors that influence mothers' wellbeing, and how to optimize access to care given the integral role mothers play in defining family dynamics, child development and wellbeing, and accessing services for the family.

## Author Contributions

**Conceptualization:** Emily Saunders, Shannon Bedford, Julie Gosselin, Nick Harris, Joshua A. Rash.

**Data curation:** Emily Saunders, Julie Gosselin.

**Formal analysis:** Emily Saunders, Joshua A. Rash.

**Methodology:** Emily Saunders, Shannon Bedford, Julie Gosselin, Nick Harris, Joshua A. Rash.

**Project administration:** Julie Gosselin.

**Supervision:** Julie Gosselin, Nick Harris, Joshua A. Rash.

**Writing – original draft:** Emily Saunders, Noah W. Pevie.

**Writing – review & editing:** Shannon Bedford, Julie Gosselin, Nick Harris, Joshua A. Rash.

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
