## [Decision Letter · Decision Letter 0]

29 Aug 2023

PONE-D-23-18643Moms in motion: Predicting healthcare utilization patterns among Mothers in Newfoundland and LabradorPLOS ONE

Dear Dr. Rash,

Thank you for submitting your manuscript to PLOS ONE. After careful consideration, we feel that it has merit but does not fully meet PLOS ONE’s publication criteria as it currently stands. Therefore, we invite you to submit a revised version of the manuscript that addresses the points raised during the review process.

We look forward to receiving your revised manuscript.

Kind regards,

Ammal Mokhtar Metwally, Ph.D (MD)

Academic Editor

PLOS ONE

3. We noted in your submission details that a portion of your manuscript may have been presented or published elsewhere. [This project constituted the doctoral dissertation of Emily Saunders. The dissertation has not been published at this time, not is this material under consideration elsewhere.] Please clarify whether this publication was peer-reviewed and formally published. If this work was previously peer-reviewed and published, in the cover letter please provide the reason that this work does not constitute dual publication and should be included in the current manuscript.

Reviewers' comments:

Reviewer's Responses to Questions

**Comments to the Author**

1. Is the manuscript technically sound, and do the data support the conclusions?

Reviewer #1: Yes

Reviewer #2: Yes

2. Has the statistical analysis been performed appropriately and rigorously? 

Reviewer #1: Yes

Reviewer #2: I Don't Know

3. Have the authors made all data underlying the findings in their manuscript fully available?

Reviewer #1: No

Reviewer #2: Yes

4. Is the manuscript presented in an intelligible fashion and written in standard English?

Reviewer #1: Yes

Reviewer #2: Yes

5. Review Comments to the Author

Reviewer #1: Hi... great article but there are few concerns

1. In the introduction kindly mention the current health care utilization data at least within 5 yrs till date.

2. How was the questionnaire validated, please state it clearly. In addition kindly explain clearly the line 220-224 supporting and justifying the same.

3. Kindly check out the data for this study for speedy usage. It might be outdated for publication usage. Since it has to be used within 5yrs after data collection.

Reviewer #2: The manuscript entitled “Moms in motion: predicting healthcare utilization patterns among mothers in Newfoundland and Labrador” is based on impressive empirical evidence and makes an original contribution. The authors have provided a lot of information about maternal and child health in this study and the data is representative of the provincial level. However, I strongly encourage the authors to address the following points.

1.Citation in this manuscript starts from number 3. Authors should begin to cite with number 1

2.Please describe introduction section using paragraphs and no need for sub headings in this section (Page 3 line 64 and Page 8 line 172)

3.There are 67 citations in introduction section out of 113 references. Authors just have only 50% citation in discussion section compared to introduction. Please more comprehensive literature review in discussion section.

4.The literature review is promising, but disregards recent publications in the field of healthcare utilization. Please more recent literature review (From 113 referenced papers in this study, 74 references are older than 2013)

5.Please clearly explain in method section regarding independent and dependent. variables. Information given in the method is a bit confusing for readers.

6.I had difficulties understanding the table 1 and table 3 and suggest that the authors simplify it. Please summarize it. If it possible, please re-categorized each variables.

6. PLOS authors have the option to publish the peer review history of their article (what does this mean?). If published, this will include your full peer review and any attached files.

Reviewer #1: No

Reviewer #2: No

---

## [Author Response · Author response to Decision Letter 0]

7 Oct 2023

Please refer to our attached document detailing or responses to reviewer concerns. Thank you.

---

## [Decision Letter · Decision Letter 1]

20 May 2024

Moms in motion: Predicting healthcare utilization patterns among Mothers in Newfoundland and Labrador

PONE-D-23-18643R1

Dear Dr. Rash,

We’re pleased to inform you that your manuscript has been judged scientifically suitable for publication and will be formally accepted for publication once it meets all outstanding technical requirements.

Kind regards,

Ammal Mokhtar Metwally, Ph.D (MD)

Academic Editor

PLOS ONE

Additional Editor Comments (optional):

Thank you for addressing all reviewers' comments. We believe your manuscript is ready for publication.

Reviewers' comments:

Reviewer's Responses to Questions

**Comments to the Author**

1. If the authors have adequately addressed your comments raised in a previous round of review and you feel that this manuscript is now acceptable for publication, you may indicate that here to bypass the “Comments to the Author” section, enter your conflict of interest statement in the “Confidential to Editor” section, and submit your "Accept" recommendation.

Reviewer #1: All comments have been addressed

Reviewer #2: All comments have been addressed

2. Is the manuscript technically sound, and do the data support the conclusions?

Reviewer #1: Yes

Reviewer #2: Yes

3. Has the statistical analysis been performed appropriately and rigorously? 

Reviewer #1: Yes

Reviewer #2: I Don't Know

4. Have the authors made all data underlying the findings in their manuscript fully available?

Reviewer #1: Yes

Reviewer #2: Yes

5. Is the manuscript presented in an intelligible fashion and written in standard English?

Reviewer #1: Yes

Reviewer #2: Yes

6. Review Comments to the Author

Reviewer #1: Moms in motion really gives credibility to mothers. The article meets scientific standard . I wish the authors the best in future endeavors.

However it would be encouraging, if carried forward and following more intervention. Thank you

Reviewer #2: (No Response)

7. PLOS authors have the option to publish the peer review history of their article (what does this mean?). If published, this will include your full peer review and any attached files.

Reviewer #1: No

Reviewer #2: No

---

## [Editor Report · Acceptance letter]

27 Jun 2024

PONE-D-23-18643R1 

PLOS ONE

Dear Dr. Rash, 

I'm pleased to inform you that your manuscript has been deemed suitable for publication in PLOS ONE. Congratulations! Your manuscript is now being handed over to our production team.

Kind regards, 

on behalf of

Professor Ammal Mokhtar Metwally 

Academic Editor

PLOS ONE